# Controlled large non-reciprocal charge transport in an intrinsic magnetic topological insulator MnBi$_2$Te$_4$

Zhaowei Zhang [1,9], Naizhou Wang[1,9], Ning Cao[2], Aifeng Wang [2], Xiaoyuan Zhou [2], Kenji Watanabe [3], Takashi Taniguchi [4], Binghai Yan [5] ✉ & Wei-bo Gao [1,6,7,8] ✉

Symmetries, quantum geometries and electronic correlations are among the most important ingredients of condensed matters, and lead to nontrivial phenomena in experiments, for example, non-reciprocal charge transport. Of particular interest is whether the non-reciprocal transport can be manipulated. Here, we report the controllable large non-reciprocal charge transport in the intrinsic magnetic topological insulator MnBi$_2$Te$_4$. The current direction relevant resistance is observed at chiral edges, which is magnetically switchable, edge position sensitive and stacking sequence controllable. Applying gate voltage can also effectively manipulate the non-reciprocal response. The observation and manipulation of non-reciprocal charge transport reveals the fundamental role of chirality in charge transport of MnBi$_2$Te$_4$, and pave ways to develop van der Waals spintronic devices by chirality engineering.

Van der Waals materials provide an interesting platform to study the intertwined magnetism and band topology[1,2], and a series of exotic states of matter emerge. Among them, quantum anomalous Hall effect (QAHE) attracted a lot of attention. For QAHE, the most interesting part would be quantized plateau of Hall conductance as well as the dissipationless chiral edge transport channels that emerge at the Chern insulator states. Besides, the chirality of the edge transport, as well as the magnetization, play an essential role in dissipative transport regimes, for instance, non-reciprocal charge transport behaviors[2–6].

Non-reciprocal response, manifesting as the resistance difference between positive and negative current, is the central process to convert an oscillating electromagnetic field to a direct current, in other words, rectification. The demand of low-power, high-frequency rectifiers inspires studies on non-reciprocal charge transport in new material systems, such as non-centrosymmetric crystals[7–12], topological insulators[13–17], magnet/superconductor interfaces[18,19], topological insulator/superconductor interfaces[20] and magnet/topological insulator interfaces[3,21,22]. Especially, a large non-reciprocal charge transport mediated by quantum anomalous Hall edge states has been observed in magnetically doped topological insulator[14]. As compared to traditional magnetically doped topological insulators, the intrinsic magnetic topological insulator MnBi$_2$Te$_4$ provides more robust QAHE, since it does not introduce magnetic impurities and remove the need of precise control of element species. Due to the ferromagnetic intralayer coupling and antiferromagnetic interlayer coupling, MnBi$_2$Te$_4$ also hosts rich magnetic phases, including fully compensated antiferromagnetic, uncompensated antiferromagnetic and spin-aligned ferromagnetic states. External magnetic field turns out to be an

[1]Division of Physics and Applied Physics, School of Physical and Mathematical Sciences, Nanyang Technological University, Singapore 637371, Singapore. [2]Low Temperature Physics Laboratory, College of Physics and Center for Quantum Materials and Devices, Chongqing University, 401331 Chongqing, China. [3]Research Center for Functional Materials, National Institute for Materials Science, 1-1 Namiki, Tsukuba 305-0044, Japan. [4]International Center for Materials Nanoarchitectonics, National Institute for Materials Science, 1-1 Namiki, Tsukuba 305-0044, Japan. [5]Department of Condensed Matter Physics, Weizmann Institute of Science, Rehovot 7610001, Israel. [6]The Photonics Institute and Centre for Disruptive Photonic Technologies, Nanyang Technological University, Singapore 637371, Singapore. [7]Centre for Quantum Technologies, National University of Singapore, Singapore, Singapore. [8]MajuLab, International Joint Research Unit, UMI 3654, CNRS, UniversitéCôte d'Azur, Sorbonne Université, National University of Singapore, Nanyang Technological University, Singapore, Singapore. [9]These authors contributed equally: Zhaowei Zhang, Naizhou Wang. ✉e-mail: binghai.yan@weizmann.ac.il; wbgao@ntu.edu.sg

effective tool to manipulate such magnetic states as well as the charge transport. Besides, the van der Waals nature of MnBi$_2$Te$_4$ makes it possible to realize more attractive phenomena by changing stacking sequence or assembling heterostructure[23–30].

In our current manuscript, we have demonstrated controllable large non-reciprocal charge transport in an intrinsic magnetic topological insulator MnBi$_2$Te$_4$. Due to its unique antiferromagnetic van-der-Waals structure, the large non-reciprocal charge transport in MnBi$_2$Te$_4$ can be manipulated by gate voltage, magnetic field and septuple layer numbers. The observation of tunable non-reciprocal resistance may help to develop multifunctional spintronic devices, such as magnetic switchable diodes and high-frequency rectification.

## Results

### Chiral edge states revealed by quantum anomalous Hall effect

First, we show the characterization of MnBi$_2$Te$_4$ devices. The device structure is shown in Fig. 1a. In details, MnBi$_2$Te$_4$ thin flakes are obtained by Al$_2$O$_3$-assisted exfoliation[31,32]. The sample thickness is determined by optical contrast as shown in Supplementary Fig. 1. We have chosen four 5-SL (Device 1, Device 2, Device 4, and Device 5) and one 4-SL (Device 3) MnBi$_2$Te$_4$ in our measurements. In the main text, we show the results from Device 1 and Device 3. Au electrodes are thermally deposited through stencil masks, which avoids the degradation from the chemical or ambient exposure and ensures the high device quality. The Si/SiO$_2$ bottom gate or the graphite/hBN top gate works as gate. We measure the longitudinal resistance of a 5-SL MnBi$_2$Te$_4$ as a function of temperature as shown in Fig. 1b. The Néel temperature is around 23 K, which is indicated by the resistance peak. The temperature dependent resistance of other two devices are shown in Supplementary Fig. 2a, b. Three devices display similar Néel temperature.

To reveal the chiral edge states in MnBi$_2$Te$_4$, we tune the Fermi level to the charge neutrality point (CNP). 5-SL MnBi$_2$Te$_4$ owns uncompensated magnetic moments under zero external magnetic field when temperature below the Néel temperature, as shown in

Fig. 1c, d. We also show the fully compensated antiferromagnetic states of a 4-SL MnBi$_2$Te$_4$ device in Supplementary Fig. 4d. We note that in the ideal quantum anomalous Hall state, only chiral edge states contribute to the charge transport and the longitudinal resistance becomes zero. However, longitudinal resistance in our MnBi$_2$Te$_4$ samples does not vanish. For our Device 2 and Device 3, the nearly quantized Hall resistance of 0.977 h/e$^2$ and 0.945 h/e$^2$ have been achieved (Supplementary Fig. 2 and Supplementary Fig. 3).

### Large non-reciprocal resistance in MnBi$_2$Te$_4$

Next, we study the non-reciprocal charge transport in MnBi$_2$Te$_4$ devices. As shown in Fig. 2a, when the current is injected in opposite directions, charge carriers at the same edge are scattered by the chiral state in different ways. This leads to a difference in resistance under current reversal, so-called non-reciprocal charge transport behaviors. Here, we adopt the phenomenological model[33] to describe the current relevant resistance:

$$V_{xx} = iR_{xx} = iR_0 + \gamma R_0 i^2 \left( \hat{M} \times \hat{P} \right) \cdot \hat{i}, \tag{1}$$

where $R_0$ is the resistance that does not change with the current, $\gamma$ is the constant that characterizes the strength of the non-reciprocal charge transport effect, $\hat{M}$ is the magnetization direction of the MnBi$_2$Te$_4$, $\hat{P}$ is the edge charge dipole which is opposite on two edges, and $\hat{i}$ is the current direction. We study the non-reciprocal charge transport by measuring the resistance difference between positive and negative currents in a 5-SL MnBi$_2$Te$_4$ as described in Fig. 2a. The 10 μA and −10 μA DC current is injected to the device respectively, and we measure the longitudinal resistance at the right edge at 11 K. A top gate voltage of −2 V is applied, at which the Fermi level is around the CNP. In Fig. 2b, we plot the resistance difference as a function of the magnetic field. The resistance difference $\Delta R$ is defined as $R_L(10 \text{ μA}) - R_L(-10 \text{ μA})$ and the value is antisymmetrized for **M**. The raw data are shown in Supplementary Fig. 5. A large resistance difference is observed, reaching the magnitude of 100 Ω. Such a resistance difference cannot originate from thermal effect as we discuss in Supplementary Note 5. We then compare the non-reciprocity in MnBi$_2$Te$_4$ and Cr-doped (Bi,Sb)$_2$Te$_3$ by $\triangle R/R_0/I$. Under 7 T magnetic field, the $\triangle R/R_0/I$ reaches 2911 A$^{-1}$ in MnBi$_2$Te$_4$, which is comparable to 2600 A$^{-1}$ in Cr-doped (Bi,Sb)$_2$Te$_3$[14].

Since the non-reciprocal response are originated from the quadratic term of current, we adopt the second-harmonic voltage measurements with small AC driven current $I^{RMS} = 5$ μA to obtain better signal-to-noise ratio and rule out other high-order effects. With the base frequency of the driven current set as 17.777 Hz, the voltage drop is written as:

$$V_{xx} = \sqrt{2}\sin(\omega t)R_0 + \gamma R_0 i^2 (1 - \cos(2\omega t)) \left( \hat{M} \times \hat{P} \right) \cdot \hat{i}. \tag{2}$$

The voltage with the frequency of 35.554 Hz reveals the strength of the non-reciprocal charge transport effect. Here, the non-reciprocal resistance is defined as $R_{xx}^{2\omega} = V_{xx}^{2\omega}/I^{RMS}$, where $V_{xx}^{2\omega}$ is the second-harmonic voltage measured by the lock-in amplifier. In Fig. 2c, we show the non-reciprocal resistance as a function of magnetic field, which is antisymmetrized for **M**. The magnitude of $\gamma$ is estimated by $\gamma = \sqrt{2}R_{xx}^{2\omega}/(R_0 \cdot I^{RMS})$. At 11 K, the $\gamma$ is $4.5 \times 10^3$ A$^{-1}$ under the magnetic field of 7 T, and $1.4 \times 10^3$ A$^{-1}$ without the external magnetic field. We also study the current magnitude dependent non-reciprocal resistance as shown in Supplementary Fig. 8. The non-reciprocal resistance scales linearly with the AC current magnitude, which agrees well with our phenomenological model.

Remarkably, the A-type antiferromagnetic order unlocks complicated magnetic states and it deeply influences the non-reciprocal resistance. Applying the magnetic field aligns magnetic moments along with the external field. By controlling the magnetization, the

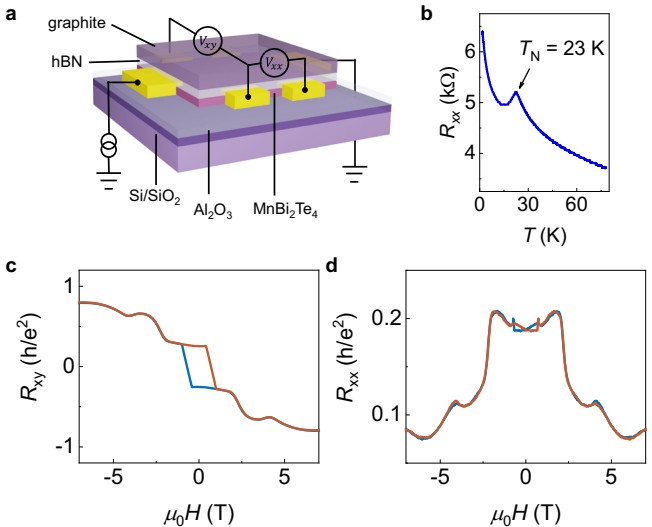

**Fig. 1 | The characterization of MnBi$_2$Te$_4$ devices. a** The schematic structure of the MnBi$_2$Te$_4$ device. The MnBi$_2$Te$_4$ thin flakes are obtained by Al$_2$O$_3$-assisted exfoliation method. Au electrodes are thermally deposited through stencil masks. Si/SiO$_2$ and graphite/hBN work as bottom and top gate, respectively. **b** The temperature-dependent resistance of a 5-SL MnBi$_2$Te$_4$ (Device 1) device. The Néel temperature ($T_N$) is indicated by a resistance peak around 23 K. **c, d** The transverse and longitudinal resistance as a function of the magnetic field at 1.7 K for Device 1. A top gate voltage of −2 V is applied to tune the Fermi energy to the charge neutrality point.

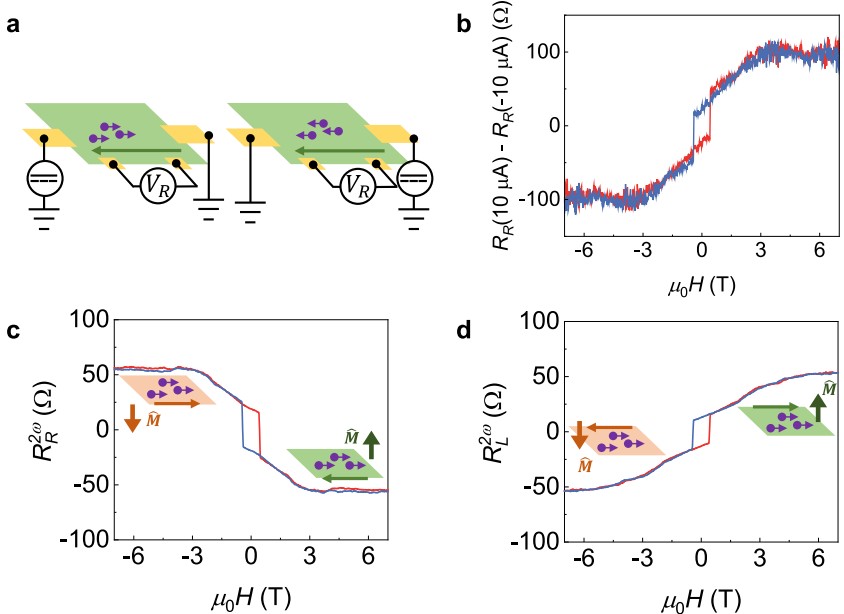

**Fig. 2 | Non-reciprocal charge transport in a 5-SL MnBi₂Te₄.** **a** Schematic illustrations of current direction dependent backscattering of edge transport. The chiral edge states are indicated by green arrows. **b** Resistance difference between positive and negative current as a function of the magnetic field. The measurement is performed at 11 K with top gate voltage of −2 V. **c, d** The non-reciprocal resistance measured at the right edge and left edge, respectively. The injected current $I^{RMS}$ is 5 μA. The measurements are performed at 11 K with top gate voltage of −2 V. Insets show schematic illustrations of the interplay between chiral edge states and 2D bulk transport channels (e.g., surface states) at different magnetic states.

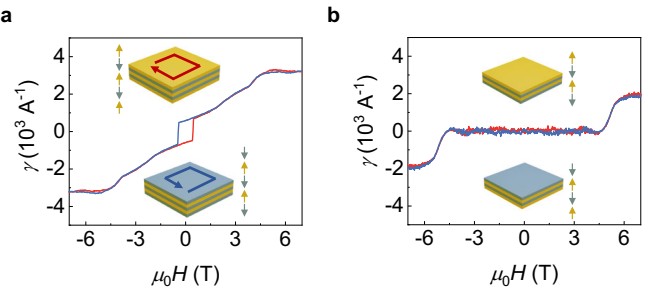

**Fig. 3 | Non-reciprocal charge transport in 5-SL MnBi₂Te₄ and 4-SL MnBi₂Te₄.** **a** Magnetic field dependent non-reciprocity for a 5-SL MnBi₂Te₄ device. A top gate voltage of 3 V is applied, and the sample is slightly electron doped. Insets show the schematic illustrations of two magnetic states under zero magnetic field. **b** Magnetic field dependent non-reciprocity for a 4-SL MnBi₂Te₄ device. The AC driven current $I^{RMS} = 1$ μA is adopted in measurements. The measurements are performed at the temperature of 11 K. A top gate voltage of 3 V is applied, and the sample is slightly electron doped. Insets show schematic illustrations of two magnetic states under zero magnetic field.

bulk/edge conduction ratio is manipulated. Meanwhile, the chirality of the edge transport can be switched by reversing magnetic moments, leading to opposite non-reciprocity at opposite magnetic states. The finite non-reciprocal resistance is observed without external magnetic field and shows the sign reversal under magnetic states ↑↓↑↓↑ and ↓↑↓↑↓ due to the uncompensated magnetic moments in the 5-SL MnBi₂Te₄. Multiple non-reciprocal resistance plateaus under magnetic field are also revealed including fully spin polarized states ↑↑↑↑↑ and ↓↓↓↓↓. The abundant magnetic field controlled non-reciprocal resistance states are absent in other non-reciprocal system, such as magnetically doped topological insulator. We show the non-reciprocal resistance in Device 2 in Supplementary Fig. 6. Apart from magnetic field, non-reciprocal resistance in MnBi₂Te₄ is edge position sensitive. In Fig. 2d, we show the non-reciprocal resistance as a function of

magnetic field measured at the left edge of the device. Non-reciprocal resistance that measured at two different edges shows the close magnitude, but the sign is reversed. This behavior indicates the broken inversion symmetry of the edge transport and the edge–charge dipole $\hat{P}$ at two edges is opposite.

Magnetic moments of 4-SL MnBi₂Te₄ are fully compensated under zero magnetic field, for which the non-reciprocal charge transport in 4-SL MnBi₂Te₄ shows different behaviors compared with that in 5-SL MnBi₂Te₄. In Fig. 3a, b, we show the field dependent non-reciprocity $\gamma$ of 5-SL and 4-SL MnBi₂Te₄ devices, respectively. The non-reciprocal resistance for both devices are measured at the right edge with slightly electron doping. We find in high-magnetic-field regime, non-reciprocity of 5-SL MnBi₂Te₄ and 4-SL exhibit the same order of magnitude. However, under zero magnetic field, non-reciprocal resistance nearly vanishes in 4-SL MnBi₂Te₄ devices. This is consistent with the phenomenological model for **M** = 0 and the current relevant term vanishes. The vanished non-reciprocity with zero magnetization can be further confirmed by measuring the non-reciprocal resistance at paramagnetic states. Increasing the temperature above the Néel temperature results in the breakdown of chiral edge states even under high magnetic field, leading to vanished non-reciprocal resistance, which is shown in Supplementary Fig. 9.

## Gate tunability of non-reciprocal charge transport

Finally, we demonstrate the gate tunability of the non-reciprocal charge transport in MnBi₂Te₄. In Fig. 4, we show the gate dependent $R_L^{2\omega}$ and $R_R^{2\omega}$ at 11 K with various top gate voltage. We find that both the magnitude and sign of the $R_{L/R}^{2\omega}$ can be tuned by applying gate voltage. In high field regime, $R_L^{2\omega}$ at the spin state ↑↑↑↑↑ is tuned from positive to negative when gate voltage is scanned from −5 V to 5 V, while $R_R^{2\omega}$ shows the opposite trend. The gate tunability is also valid under zero magnetic field, where the loop chirality is switched while scanning top gate voltage from −5 to 5 V. We note the gate voltage does not switch the ↑↓↑↓↑ and ↓↑↓↑↓ states as we discussed in Supplementary Note 9. Besides, optical magnetic circular dichroism (MCD) reveals the magnetization is gate voltage independent in the field regime of −1 to

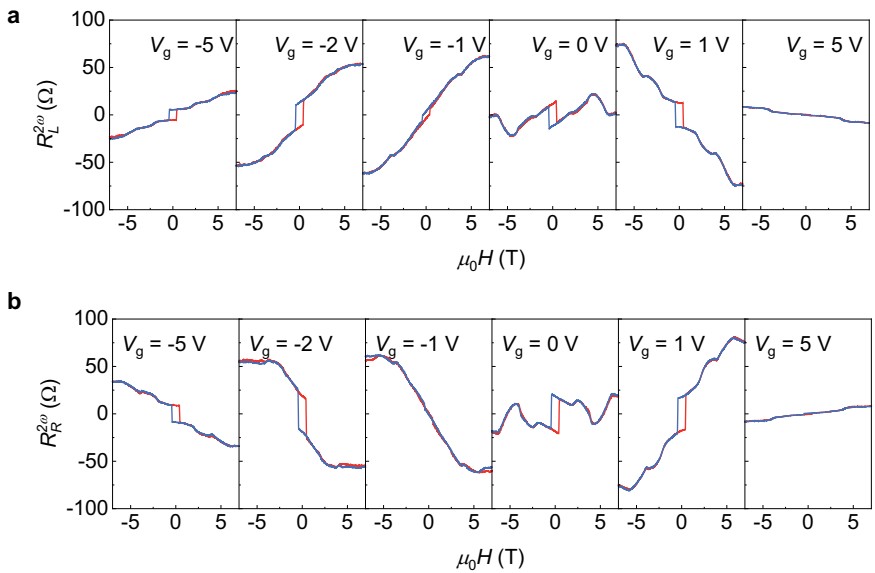

**Fig. 4 | Non-reciprocal resistance at various top gate voltages. a** Non-reciprocal resistance as a function of magnetic field measured at the left edge. **b** Non-reciprocal resistance as a function of magnetic field measured at the right edge.

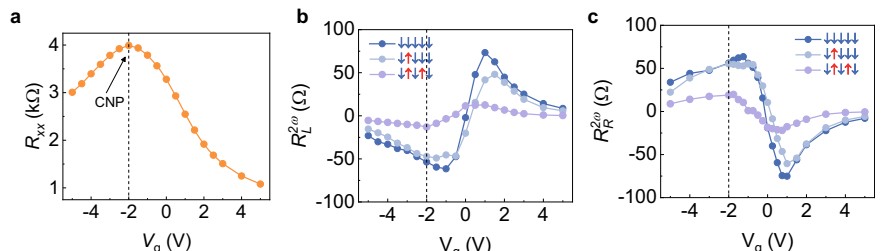

**Fig. 5 | Gate tunability of non-reciprocal resistance at different magnetic states in 5-SL MnBi₂Te₄. a** Longitudinal resistance of the 5-SL MnBi₂Te₄ as a function of the top gate voltage under zero magnetic field. The measurement is carried out at 11 K. The resistance shows a maximum at the charge neutrality point (CNP) with the top gate voltage of −2 V. **b**, **c** Non-reciprocal resistance measured at left edge and right edge of the 5-SL MnBi₂Te₄, respectively. Different curves show the non-reciprocal resistance at spin states of ↓↓↓↓↓, ↓↑↓↓↓ and ↓↑↓↑↓.

1 T. Therefore, it is the change of Fermi energy that affects the non-reciprocal charge transport.

To provide more insights into the effect of gate voltage on the non-reciprocal charge transport, we summarize the gate dependence of longitudinal resistance and non-reciprocal resistance in Fig. 5. As shown in Fig. 5a, the longitudinal resistance under zero magnetic field shows the maximum at the top gate voltage of −2 V. We also show sign reversal of the $R_L^{2\omega}$ and $R_R^{2\omega}$ while scanning gate voltage at different magnetic states in Fig. 4b, c. Further study about magnetic-field-direction dependent non-reciprocal charge transport is shown in Supplementary Fig. 13. A dip of $R_R^{2\omega}$ when the $R_{xy}$ reaches the maximum value under an out-of-plane magnetic field is observed in Device 4, indicating that the coexistence of chiral edge transport and bulk transport plays an important role in the non-reciprocal charge transport.

## Discussion

The remarkable features of non-reciprocal resistance can be rationalized by a simple band structure scenario on the QAHE edge. At the CNP, we should note that our devices deviate slightly from the fully insulating state and $R_{xy}$ being smaller than $h/e^2$ (see Fig. 1c). Therefore, one chiral edge state and trivial edge states (including subbands from the 2D bulk) coexist, from which the local edge transport is detected by two voltage electrodes. Apart from MnBi₂Te₄, prior studies on

magnetically doped topological insulators show the small finite longitudinal resistance in presence of nonchiral edge states and residual bulk states[34]. In the ideal QAHE case, the pure chiral edge state would not lead to non-reciprocal transport, since its $R_{xx} = 0$. In the weakly doped QAHE which is the case in present work, chiral edge state hybridizes strongly with trivial edge states, leading to asymmetric dispersion between opposite momenta, i.e., different magnitudes of Fermi velocities along opposite directions. This velocity asymmetry coincides with the fact that both inversion symmetry and time-reversal symmetry are broken on the edge. Suppose a finite relaxation time, the direction-dependent mean free path comes with direction-dependent Fermi velocity and eventually leads to direction-dependent resistance. Moreover, the induced resistance change is maximized near the band edge of trivial states, explaining the large non-reciprocal effect near the CNP[6]. Further, the velocity asymmetry flips order between conduction and valence bands, or when reversing the magnetism or changing edge sides, rationalizing related experimental sign changes.

In summary, we have demonstrated the non-reciprocal charge transport in an intrinsic magnetic topological insulator MnBi₂Te₄. The broken inversion symmetry and broken time-reversal symmetry in the dissipative charge transport regime are key components of the non-reciprocity in our study. By manipulating the symmetry breaking, the septuple layer dependent non-reciprocity turns out to be magnetically controllable and edge position sensitive. Meanwhile, the observed

non-reciprocal resistance can be tuned by gate voltage. We ascribe the observed non-reciprocal resistance to the interaction between chiral edge states and dissipative states. Our finding paves the way to build next-generation spintronic devices though chirality engineering.

## Methods

### Device fabrication

High-quality $MnBi_2Te_4$ crystals are grown by flux methods. $MnBi_2Te_4$ thin flakes are obtained by $Al_2O_3$ assisted exfoliation method. We determine the thickness of $MnBi_2Te_4$ thin flakes by optical contrast of captured optical images of $MnBi_2Te_4/Al_2O_3$ on PDMS stamps. $MnBi_2Te_4$ thin flakes, as well as the $Al_2O_3$ are then transferred on $Si/SiO_2$ substrates. Au electrodes are thermally deposited through stencil masks. Finally, the hBN and graphite are transferred on top of the selected $MnBi_2Te_4$ flake. The fabrication processes before covering hBN and graphite are performed in a $N_2$-filled glove box.

### Electrical measurements

DC transport measurements are performed by a current source (Keithley 6221) and a nanovoltmeter (Keithley 2182A). The top gate voltage and bottom gate voltage are applied by a dual-channel sourcemeter (Keithley 2636B). AC transport measurements are performed by a current source (Keithley 6221) and a lock-in amplifier (Zurich MFLI). All measurements are carried out with a Cryomagnetic cryostat.

## Data availability

The data that support the findings of this study are available from the corresponding authors upon reasonable request.

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

## Acknowledgements

This work is supported by the Singapore National Research Foundation through its Competitive Research Program (CRP Award No. NRF-CRP22-2019-0004 and Quantum engineering program) and Singapore Ministry of Education (MOE2016-T3-1-006 (S)). B.Y. acknowledges the financial support by the European Research Council (ERC Consolidator Grant "NonlinearTopo," No. 815869) and the ISF - Personal Research Grant (No. 2932/21). Work at Chongqing University was financially supported by the National Natural Science Foundation of China (Grants No. 12004056, No. 52071041). K.W. and T.T. acknowledge support from the JSPS KAKENHI (Grant Numbers 19H05790, 20H00354 and 21H05233).

## Author contributions

W.-b.G., and B.Y. conceived the experiment. N.C., A.W., and X.Z. synthesized the $MnBi_2Te_4$ crystal. T.T. and K.W. provided the hexagonal

boron nitride samples. Z.Z. and N.W. fabricated the $MnBi_2Te_4$ devices and carried out the electrical measurements. B.Y. and W.-b.G. performed theoretical analysis. Z.Z., N.W., B.Y, and W.-b.G wrote the manuscript with extensive input from the other authors. W.-b.G. and B.Y. supervised the project.

## Competing interests

The authors declare no competing interests.
