## [Peer Review File · Nature Communications]

Reviewers' Comments:

Reviewer #1:

Remarks to the Author:

This work reports chiral edges related-non-reciprocal charge transport in an intrinsic magnetic topological insulator MnBi_2Te_4 . The non-reciprocal response is magnetically controllable, voltage gate tunable, and edge position sensitive. However, some results are not provided and the underlying physics should be discussed in more detail. The authors should address and clarify the below points before I could make the recommendation.

1. In terms of novelty, the author shows the non-reciprocal charge transport in MBT, which is an intrinsic magnetic topological insulator. Nature Nanotechnology 15, 831 (2020) has presented the non-reciprocal charge transport in the Cr-doped $(\text{Bi,Sb})_2\text{Te}_3$, which is also a magnetic topological insulator, with a comprehensive discussion on the properties and mechanism of this behavior. Considering both materials belong to the quantum anomalous Hall family, the author should clearly state the difference between the two works, as well as the novelty of this work.
2. Experimentally, the authors demonstrated that non-reciprocal charge transport in MBT was magnetically controllable, voltage gate tunable, and edge position sensitive. The physics underlying the tunable nonreciprocity should be discussed in more detail.
3. As for the origin of the non-reciprocal resistance, the Joule heating somehow leads to a temperature gradient across the sample. It can drive a thermoelectric voltage. The thermoelectrical effect could result in the nonlinearity of the I-V curve and the second-harmonic resistance. The author should provide evidence to exclude the thermoelectrical effect.
4. A large non-reciprocal charge transport is stated in the title. Non-reciprocal voltage is dependent on the applied current. It seems that a charge current of $10\ \mu\text{A}$ was applied in the dc experiment [Figure 2b]. How about the applied current in the AC experiment [Figures 2c and d]? Please show and discuss the current magnitude dependence of the non-reciprocal resistance. In addition, the author uses a phenomenological model [Eq. 1] to describe the non-reciprocal resistance. However, the quantitative analysis of γ is lacking.
5. The author should give a comprehensive comparison between MBT and other non-reciprocal charge transport systems.
6. In Figure 2b, the difference between positive and negative current is presented as a function of the applied magnetic field. For a clear comparison, it is suggested that the originated results before subtraction are added in Supplementary Information.
7. Non-reciprocal resistance in Figures 2c and d should be clearly defined based on the ac harmonic measurement.
8. In Figure 3, the author compared the non-reciprocal resistance in 4-SL and 5-SL MBT. However, according to Eq.1, the second-harmonic resistance is relevant to the longitudinal resistance R_0 . How about the longitudinal resistance R_0 of 4-SL and 5-SL? Does the author take R_0 into account when comparing the non-reciprocity in two samples?

Reviewer #2:

Remarks to the Author:

This is an interesting work on non-reciprocal charge transport phenomena in an intrinsic magnetic topological insulator MnBi_2Te_4 , whose unique antiferromagnetic van-der-Waals nature allows for the controllable large non-reciprocal resistance by gate voltage, magnetic field and layer numbers. This paper experimentally shows how promising for tunable non-reciprocity the use of the intrinsic magnetic topological insulator MnBi_2Te_4 is. I think that this study brings new insights into developing magnetic switchable diodes and high-frequency rectifiers based on van-der-Waals materials.

I would recommend this paper for publication in Nature Communications if the following questions are properly answered.

- 1) It would be great if the authors could make a quantitative comparison of the diode efficiency obtained from MnBi_2Te_4 devices with a previous achievement (26%) of Cr-doped $(\text{Bi,Sb})_2\text{Te}_3$ [Nature Nanotechnology 15, 831 (2020)].

2) As was done in the previous study on Cr-doped (Bi,Sb)₂Te₃ [Fig. 3, Nature Nanotechnology 15, 831 (2020)], the authors need to investigate how the magnetic-field-direction dependence of the non-reciprocal resistance looks like and to compare it with that of the Hall resistance. I think that these data are useful for explaining better the gate dependence and identifying what the underlying mechanism is. I am also curious how fundamentally different the suggested mechanism of non-reciprocity in this study is from the interplay between chiral edge state and surface state proposed previously [Nature Nanotechnology 15, 831 (2020)].

3) Although the gate voltage does not switch the $\uparrow\uparrow\downarrow\downarrow$ to the $\downarrow\downarrow\uparrow\uparrow$ state (Fig. 4), there does exist a strong suppression of perpendicular magnetic anisotropy (by a factor of around 5, Fig. S8) when applying the gate voltage from -5 to +4 V. I agree with the authors that the sign change of the non-reciprocal resistance comes from the change of Fermi energy (relative to CNL). Yet, the gate-voltage-dependent magnetic anisotropy can also contribute to the magnitude. I think that the authors should clarify this point to avoid confusion.

I hope it helps improve the overall quality of the paper to be more suited for the high-impact journal of Nature Communications.

Reviewer #3:

Remarks to the Author:

The manuscript entitled "Controlled large non-reciprocal charge transport in an intrinsic magnetic topological insulator MnBi₂Te₄" by Z. Zhang et al., described the nonreciprocal transport in magnetic topological insulator MnBi₂Te₄. They reported gate- and magnetic-field-tunable nonreciprocal resistance. The authors attributed it to the hybridization between chiral edge state and trivial edge state. Data are nicely shown. However, there are a few issues which need to be addressed in this manuscript before it is suitable to be published in Nature Communications.

1. In the previous studies, nonreciprocal transport in MnBi₂Te₄ (C. Ye et al., Nano letters 22, 1366-1373 (2022)) and gate-controlled nonreciprocity in magnetic topological insulator (K. Yasuda et al., Physical Review Letters 117, 127202 (2016) and K. Yasuda et al., Nature Nanotechnology 15, 831-835 (2020)) has been already reported. What is the novelty of this paper?

2. Why hysteresis behavior in nonreciprocal signals at $V_g = -1$ (only R) and 5 V disappears (Fig. 4)? In that condition, hysteresis in R_{yx} is observed? For the clear understanding of the magnetic properties, you can show R_{yx} simultaneously in Fig.4. BTW, are there intermediate state in 4SL samples such as $\uparrow\uparrow\downarrow\downarrow$ or $\downarrow\downarrow\uparrow\uparrow$?

3. The CNP is different between 5SL and 4SL samples (Fig. S3). Does this come from the sample dependence or layer number dependence? BTW, Fig. S3 is not mentioned in the main text nor Supplementary Materials.

4. There are several minor comments:

(a) In the citation list, article numbers are sometimes wrong especially for Nature journals.

(b) As for the nonreciprocal transport, there are other previous works:

R. Wakatsuki et al., Sci. Adv. 3 e160239 (2017).

P. He et al., PRL 120 266802 (2018)

J. Lustikova et al., Nature Communications 9 4922 (2018).

Y. M. Itahashi et al., Sci. Adv. 6 eaay9120 (2020).

Response to Reviewers' comments

Reviewer #1 (Remarks to the Author):

This work reports chiral edges related-non-reciprocal charge transport in an intrinsic magnetic topological insulator MnBi_2Te_4 . The non-reciprocal response is magnetically controllable, voltage gate tunable, and edge position sensitive. However, some results are not provided and the underlying physics should be discussed in more detail. The authors should address and clarify the below points before I could make the recommendation.

Comment #1.1: In terms of novelty, the author shows the non-reciprocal charge transport in MBT, which is an intrinsic magnetic topological insulator. Nature Nanotechnology 15, 831 (2020) has presented the non-reciprocal charge transport in the Cr-doped $(\text{Bi,Sb})_2\text{Te}_3$, which is also a magnetic topological insulator, with a comprehensive discussion on the properties and mechanism of this behavior. Considering both materials belong to the quantum anomalous Hall family, the author should clearly state the difference between the two works, as well as the novelty of this work.

Response #1.1: We sincerely thank the reviewer for such a great question. We are grateful to have an opportunity to discuss the difference between MnBi_2Te_4 and Cr-doped $(\text{Bi,Sb})_2\text{Te}_3$ and the novelty of our work. The non-reciprocal charge transport behaviors are usually observed in noncentrosymmetric quantum materials when the time-reversal symmetry is further broken. A unique property of quantum anomalous Hall materials is the conducting edge with the insulating bulk originating from the intertwined magnetism and band topology. When the charge transport is in the dissipative regime, this scenario supports the non-reciprocal charge transport because the chiral edge transport breaks the inversion symmetry and the magnetic moment breaks the time-reversal symmetry. In this sense, MnBi_2Te_4 and Cr-doped $(\text{Bi,Sb})_2\text{Te}_3$ fall into similar material systems. In fact, MnBi_2Te_4 is a new member of the quantum anomalous Hall family and many fundamental questions remains. It is desired to understand the charge transport in MnBi_2Te_4 , and study potential applications regarding following aspects:

- (1) Intrinsic magnetic topological insulator: For magnetically doped topological insulators, to achieve quantum anomalous Hall effect, the ratio of elements in Cr-doped $(\text{Bi,Sb})_2\text{Te}_4$ needs to be precisely controlled, which is technically difficult. Meanwhile, the sample quality is fundamentally limited due to magnetic dopants. However, this obstacle is removed in the intrinsic magnetic topological insulator MnBi_2Te_4 , which ensures the high sample quality.
- (2) Multiple magnetic states resulting from the antiferromagnetic interlayer coupling: MnBi_2Te_4 is an A-type antiferromagnet. Within the SL, magnetic moments are ferromagnetically coupled. Magnetic moments in two neighboring SL are antiferromagnetically coupled. Such magnetic structures provide us with new degree of freedoms to manipulate the charge transport. First, without external magnetic field, odd-SL-number MnBi_2Te_4 is an uncompensated antiferromagnet with nonzero net magnetic moments, and even-SL-number MnBi_2Te_4 is a fully compensated antiferromagnet with zero net magnetic moments. Second, applying external magnetic field to multi-SL-number MnBi_2Te_4 aligns magnetic moments through spin-flip and/or spin-flop transitions. In contrast to ferromagnetic Cr-doped $(\text{Bi,Sb})_2\text{Te}_3$, magnetic phases of antiferromagnetic MnBi_2Te_4 are much richer, and thus host new possibilities to manipulate the charge transport in different magnetic states.
- (3) Capability of building van der Waals spintronics: While designing spintronic devices, van der Waals heterostructures are attractive. The atomically flat surface and the sharp interface are crucial to achieve high performance. The van-der-Waals nature of Mn_2BiTe_4 enables it to be combined with other van-der-Waals materials and achieve new device functions.

Based on above discussions, we believe our work provides new insights into the charge transport in MnBi_2Te_4 . We hope the reviewer found our reply reasonable and thorough.

According to the comments, we have added “Due to the ferromagnetic intralayer coupling and antiferromagnetic interlayer coupling, MnBi₂Te₄ also hosts rich magnetic phases, including fully compensated antiferromagnetic, uncompensated antiferromagnetic and spin-aligned ferromagnetic states. External magnetic field turns out to be an effective tool to manipulate such magnetic states as well as the charge transport.” into the **Introduction** (page 2 of the main text), and “The broken inversion symmetry and broken time-reversal symmetry in the dissipative charge transport regime are key components of the non-reciprocity in our study. By manipulating the symmetry breaking, the septuple layer dependent non-reciprocity turns out to be magnetically controllable and edge position sensitive.” into the **Discussion** (page 9 of the main text).

Comment #1.2: Experimentally, the authors demonstrated that non-reciprocal charge transport in MBT was magnetically controllable, voltage gate tunable, and edge position sensitive. The physics underlying the tunable nonreciprocity should be discussed in more detail.

Response #1.2: We thank the reviewer for the important comment. We totally agree with the reviewer that the physics underlying the tunable non-reciprocity should be clearly discussed. In our study, the key component of the non-reciprocal charge transport is the coexistence of broken inversion symmetry and broken time-reversal symmetry in the dissipative charge transport regime. MnBi₂Te₄ is an intrinsic magnetic topological insulator with ferromagnetic intralayer coupling and antiferromagnetic interlayer coupling. The time-reversal symmetry is spontaneously broken when the temperature is below its Néel temperature. The chiral edge transport breaks the inversion symmetry. To describe the roles of broken inversion symmetry and time-reversal symmetry, we adopt a phenomenological model: $V_{xx} = iR_{xx} = iR_0 + \gamma R_0 i^2 (\hat{M} \times \hat{P}) \cdot \hat{i}$, where R_0 is the resistance that does not change with the current, γ is the constant that characterizes the strength of the non-reciprocal charge transport effect, \hat{M} is the magnetization direction of the MnBi₂Te₄, \hat{P} is the edge charge dipole which is opposite on two edges, and \hat{i} is the current direction. In this model, \hat{M} term breaks the time-reversal symmetry and \hat{P} term breaks the inversion symmetry. The tunability of the non-reciprocity relies on controlling the symmetry breaking. In the following, we would like to discuss the effect of magnetic field, gate voltage, and edge position on the non-reciprocal charge transport.

- (1) Magnetic field plays roles in manipulating magnetic states and edge transport chirality. Applying the magnetic field aligns magnetic moments along with the external field. By controlling the magnetization, the bulk/edge conduction ratio is manipulated and thus changes the non-reciprocal resistance. Meanwhile, the chirality of the edge transport can be switched by reversing magnetic moments, leading to opposite non-reciprocity at opposite magnetic states.
- (2) Applying the gate voltage changes the location of Fermi level. In heavily electron/hole doped regime, the bulk channel dominates the charge transport, in which the inversion symmetry is conserved, leading to weak non-reciprocity. When tuning the Fermi level near the CNP, the non-reciprocal charge transport reaches the maximum magnitude because of the coexistence of chiral edge transport channels and bulk transport channels, where inversion symmetry and time-reversal symmetry are broken on the edge. The gate voltage switchable non-reciprocity can be ascribed to the flipping order of the Fermi velocity asymmetry between conduction and valence bands.
- (3) The edge charge dipole is opposite between left edge and right edge, and thus the non-reciprocal charge transport shows opposite behaviours on two edges.

According to the comments, we have revised the manuscript by including “Applying the magnetic field aligns magnetic moments along with the external field. By controlling the magnetization, the bulk/edge conduction ratio is manipulated. Meanwhile, the chirality of the edge transport can be switched by reversing magnetic moments, leading to opposite non-reciprocity at opposite magnetic states.” into the **Large non-reciprocal resistance in MnBi₂Te₄** (page 5 of the main text), and “The broken inversion symmetry and broken time-reversal symmetry in the dissipative charge transport regime are key components of the non-reciprocity in our study. By manipulating the symmetry breaking, the septuple

layer dependent non-reciprocity turns out to be magnetically controllable and edge position sensitive.” into the **Discussion** (page 9 of the main text).

Comment #1.3 As for the origin of the non-reciprocal resistance, the Joule heating somehow leads to a temperature gradient across the sample. It can drive a thermoelectric voltage. The thermoelectrical effect could result in the nonlinearity of the I-V curve and the second-harmonic resistance. The author should provide evidence to exclude the thermoelectrical effect.

Response #1.3: We thank the reviewer for the insightful comment. We agree with the reviewer that the Joule heating might lead to second-harmonic resistance, which should be seriously considered. Let us consider the thermal gradient ∇T that develops in the sample. It will result in a voltage perpendicular to both the thermal gradient and the magnetization direction. However, the second harmonic resistance resulting from the Nernst effect reverses the sign when the Fermi level is across the charge neutrality. In contrast to Nernst effect, our data show a large second-harmonic resistance at the charge neutrality. We also carefully estimated the non-reciprocal resistance originating from Nernst effect. The reported Nernst coefficient of MnBi_2Te_4 under 9 T out-of-plane magnetic field is less than $1 \mu\text{V/K}$ [Zhang, H., et al. Physical Review B 105, 184411 (2022)]. At 10 K, when the injected current is $10 \mu\text{A}$, the clear hysteresis loops indicate that the sample temperature is below its Néel temperature 23 K. This means temperature increase due to the Joule heating is smaller than 13 K. The maximum resistance difference between $R(-10 \mu\text{A})$ and $R(10 \mu\text{A})$ resulting from Nernst effect should be 2.6Ω . The experimental observation shown in Figure 2(b) in the main text is 100Ω , which is much larger than that we could expect from Nernst effect. Therefore, the non-reciprocal resistance observed in MnBi_2Te_4 cannot be attributed to the thermoelectrical effect.

We have now included the above discussion into SI. We apologize for the confusion caused and we hope the reviewer finds our reply reasonable.

Comment #1.4: A large non-reciprocal charge transport is stated in the title. Non-reciprocal voltage is dependent on the applied current. It seems that a charge current of $10 \mu\text{A}$ was applied in the dc experiment [Figure 2b]. How about the applied current in the AC experiment [Figures 2c and d]? Please show and discuss the current magnitude dependence of the non-reciprocal resistance. In addition, the author uses a phenomenological model [Eq. 1] to describe the non-reciprocal resistance. However, the quantitative analysis of γ is lacking.

Response #1.4: Thank you for the important questions and suggestions. We apologize for the unclear presentation. We would like to reply comment #1.4 point by point as follows:

- (1) In AC experiments that we presented in Figure 2c and d, the current I^{RMS} is $5 \mu\text{A}$. We have now included it into the main text and added a sentence in the caption of Figure 2: “The injected current I^{RMS} is $5 \mu\text{A}$.” (page 6 of the main text)
- (2) Regarding the current magnitude dependence of the non-reciprocal resistance, we measured non-reciprocal resistances at various current magnitude. We show the data in Figure R1.1. The non-reciprocal resistance scales linearly with the injected AC current. The linear relationship is consistent with our phenomenological model (Eq1). We have now included “We also study the current magnitude dependent non-reciprocal resistance as shown in Fig. S8. The non-reciprocal resistance scales linearly with the AC current magnitude, which agrees well with our phenomenological model.” into the **Large non-reciprocal resistance in MnBi_2Te_4** (page 5 of the main text) and included the Figure R1.1 into the SI.

Figure R1.1: Current magnitude dependent non-reciprocal resistance of S4. All measurements are performed at 10 K. The sample is slightly hole doped. **a**, Out-of-plane magnetic field dependent non-reciprocal resistance with various current. All data are antisymmetrized. **b**, **c** and **d**, Current magnitude dependent non-reciprocal resistance at the magnetic state $\uparrow\downarrow\uparrow\downarrow$, $\uparrow\downarrow\downarrow\downarrow$, and $\downarrow\downarrow\downarrow\downarrow$.

- (3) Regarding the value of γ , we have now included “The magnitude of γ is estimated by $\gamma = \sqrt{2}R_{xx}^{2\omega}/(R_0 \cdot I^{RMS})$. At 11 K, the γ is $4.5 \times 10^3 A^{-1}$ under the magnetic field of 7 T, and $1.4 \times 10^3 A^{-1}$ without the external magnetic field.” **Large non-reciprocal resistance in MnBi₂Te₄** (page 5 of the main text).

Comment #1.5: The author should give a comprehensive comparison between MBT and other non-reciprocal charge transport systems.

Response #1.5: Thank you for the important suggestion. Non-reciprocal charge transport behaviors have been observed in many material systems, such as non-centrosymmetric crystals, topological insulators, magnet/superconductor interfaces, topological insulator/superconductor interfaces and magnet/topological insulator interfaces. Meanwhile, recent studies also show non-reciprocal charge transport in twisted moiré heterostructures [Lin, J. X., et al. arXiv:2112.07841 (2021)]. Compared to material systems mentioned above, MnBi₂Te₄ shows some unique properties:

- (1) Multiple non-reciprocal resistance states originating from the antiferromagnetic interlayer coupling. MnBi₂Te₄ is an A-type antiferromagnet, where magnetic moments are ferromagnetically coupled within the SL and antiferromagnetically coupled between two neighboring SLs. For a 5-SL MnBi₂Te₄, applying an out-of-plane magnetic field results in various magnetic states, such as $\uparrow\downarrow\uparrow\downarrow$, \uparrow , $\downarrow\downarrow\downarrow\downarrow$, $\uparrow\uparrow\uparrow\uparrow$, $\downarrow\downarrow\downarrow\downarrow$, $\uparrow\uparrow\uparrow\uparrow$, and $\downarrow\downarrow\downarrow\downarrow$. Tuning the magnetization of the sample effectively modifies bulk-to-edge conduction ratio, manifesting in the Hall resistance plateaus and non-reciprocal resistance plateaus.
- (2) Layer-number manipulation of the non-reciprocal charge transport. The antiferromagnetic interlayer coupling makes even-SL and odd-SL MnBi₂Te₄ magnetically different under zero magnetic field. Without external magnetic field, odd-SL-number MnBi₂Te₄ is an uncompensated antiferromagnet with nonzero net magnetic moments, and even-SL-number MnBi₂Te₄ is a fully compensated antiferromagnet with zero net magnetic moments. Because of the different magnetic states, 5-SL samples show nonzero non-reciprocal resistance but 4-SL samples show almost vanished non-reciprocal resistance without magnetic field.

In summary, as an intrinsic magnetic topological insulator, MnBi_2Te_4 provides a new platform to study the non-reciprocal charge transport. The effective manipulation of the non-reciprocity also paves ways to design functional spintronic devices.

Comment #1.6: In Figure 2b, the difference between positive and negative current is presented as a function of the applied magnetic field. For a clear comparison, it is suggested that the originated results before subtraction are added in Supplementary Information.

Response #1.6: We sincerely thank the reviewer for the suggestion. We totally agree that it will be much clear to show original and antisymmetrized data here.

Figure R1.2: Raw data and antisymmetrized data measured with DC current of Device 1. The measurement temperature is 11 K, with a top gate voltage of -2 V. (a) Raw data of the magnetic field dependence of the resistance difference between $R_R(10 \mu A)$ and $R_R(-10 \mu A)$. (b) Antisymmetrized data of the magnetic field dependence of the resistance difference between $R_R(10 \mu A)$ and $R_R(-10 \mu A)$.

We have now included “The raw data is shown in Fig. S5.” into the **Large non-reciprocal resistance in MnBi_2Te_4** (page 5 of the main text), and included the Figure R1.2 into the SI.

Comment #1.7: Non-reciprocal resistance In Figures 2c and d should be clearly defined based on the ac harmonic measurement.

Response #1.7: We sincerely thank the reviewer for pointing out the unclear definition, which is very helpful to improve the article. We have included the definition of the non-reciprocal resistance into the **Large non-reciprocal resistance in MnBi_2Te_4** (page 5 of the main text):

“Here, the non-reciprocal resistance is defined as $R_{xx}^{2\omega} = V_{xx}^{2\omega} / I^{RMS}$, where $V_{xx}^{2\omega}$ is the second-harmonic voltage measured by the lock-in amplifier.”

Comment #1.8: In Figure 3, the author compared the non-reciprocal resistance in 4-SL and 5-SL MBT. However, according to Eq.1, the second-harmonic resistance is relevant to the longitudinal resistance R_0 . How about the longitudinal resistance R_0 of 4-SL and 5-SL? Does the author take R_0 into account when comparing the non-reciprocity in two samples?

Response #1.8: Thank you for the insightful suggestion. We agree with the reviewer and have now used γ to compare the non-reciprocal charge transport in 5-SL and 4-SL MnBi_2Te_4 . The γ is calculated by $\gamma = \sqrt{2}R_{xx}^{2\omega} / (R_0 \cdot I^{RMS})$, where $R_{xx}^{2\omega}$ is defined as $R_{xx}^{2\omega} = V_{xx}^{2\omega} / I^{RMS}$. To clearly show how we obtain the value of γ , we also show the out-of-plane magnetic field dependent longitudinal resistance and non-reciprocal resistance in Figure R1.4. In the high-magnetic-field regime, magnetic moments are aligned in one direction, and non-reciprocal charge transports in 5-SL and 4-SL MnBi_2Te_4 show very similar behaviors with close magnitude of γ . Without external magnetic field, odd-SL MnBi_2Te_4 is a fully compensated antiferromagnet with vanished non-reciprocity. This agrees well the

phenomenological model when net magnetization becomes zero. We have now revised the Figure 3 in the main text and included Figure R1.4 in SI.

Figure R1.3: Non-reciprocal charge transport in 5-SL MnBi_2Te_4 and 4-SL MnBi_2Te_4 . **a**, Magnetic field dependent non-reciprocity for a 5-SL MnBi_2Te_4 device. A top gate voltage of 3 V is applied, and the sample is slightly electron doped. Insets show the schematic illustrations of two magnetic states under zero magnetic field. **b**, Magnetic field dependent non-reciprocity for a 4-SL MnBi_2Te_4 device. The AC driven current $I^{\text{RMS}} = 1 \mu\text{A}$ is adopted in measurements. The measurements are performed at the temperature of 11 K. A top gate voltage of 3 V is applied, and the sample is slightly electron doped. Insets show schematic illustrations of two magnetic states under zero magnetic field.

Figure R1.4: Comparison of the non-reciprocal resistance in 5-SL (S1) and 4-SL (S3) MnBi_2Te_4 . For the 5-SL MnBi_2Te_4 , the applied current I^{RMS} is $5 \mu\text{A}$ with a top gate voltage of 3 V. For the 4-SL MnBi_2Te_4 , the applied current is $5 \mu\text{A}$ with a top gate voltage of 3V. **a**, Out-of-plane magnetic field dependent longitudinal resistance of the 5-SL MnBi_2Te_4 (Device 1). **b**, Out-of-plane magnetic field dependent non-reciprocal resistance of the 5-SL MnBi_2Te_4 (Device 1). **c**, Out-of-plane magnetic field dependent longitudinal resistance of the 4-SL MnBi_2Te_4 (Device 3). **d**, Out-of-plane magnetic field dependent non-reciprocal resistance of the 4-SL MnBi_2Te_4 (Device 3).

Reviewer #2 (Remarks to the Author):

This is an interesting work on non-reciprocal charge transport phenomena in an intrinsic magnetic topological insulator MnBi_2Te_4 , whose unique antiferromagnetic van-der-Waals nature allows for the controllable large non-reciprocal resistance by gate voltage, magnetic field and layer numbers. This paper experimentally shows how promising for tunable non-reciprocity the use of the intrinsic magnetic topological insulator MnBi_2Te_4 is. I think that this study brings new insights into developing magnetic switchable diodes and high-frequency rectifiers based on van-der-Waals materials.

I would recommend this paper for publication in Nature Communications if the following questions are properly answered.

Comment 2.1: It would be great if the authors could make a quantitative comparison of the diode efficiency obtained from MnBi_2Te_4 devices with a previous achievement (26%) of Cr-doped $(\text{Bi,Sb})_2\text{Te}_3$ [Nature Nanotechnology 15, 831 (2020)].

Response #2.1: We sincerely thank the reviewer for the great suggest. We completely agree with the reviewer that the quantitative comparison of non-reciprocal charge transport between MnBi_2Te_4 and Cr-doped $(\text{Bi,Sb})_2\text{Te}_3$ is necessary. The $\Delta R/R_0$ reaches 26% at 100 μA in Cr-doped $(\text{Bi,Sb})_2\text{Te}_3$, which shows a high diode efficiency [Nature Nanotechnology 15, 831 (2020)]. It should be noted that ΔR scales linearly with the current magnitude, and R_0 is a constant as changing the current magnitude. Therefore, the magnitude of $\Delta R/(IR_0)$ is a better parameter to compare the non-reciprocity among different materials. The $\Delta R/(IR_0)$ in Cr-doped $(\text{Bi,Sb})_2\text{Te}_3$ is 2600 A^{-1} . In our measurements, the current is 10 μA . The R_0 is 3435 Ω and ΔR is 100 Ω under 7 T magnetic field, The $\Delta R/(IR_0)$ reaches 2911 A^{-1} , which is comparable to that in Cr-doped $(\text{Bi,Sb})_2\text{Te}_3$. Therefore, we demonstrate that non-reciprocal charge transport in MnBi_2Te_4 is large.

We have now included “We then compare the non-reciprocity in MnBi_2Te_4 and Cr-doped $(\text{Bi,Sb})_2\text{Te}_3$ by $\Delta R/R_0/I$. Under 7 T magnetic field, the R_0 is 3435 Ω and ΔR is 100 Ω . The $\Delta R/R_0/I$ reaches 2911 A^{-1} in MnBi_2Te_4 , which is comparable to 2600 A^{-1} in Cr-doped $(\text{Bi,Sb})_2\text{Te}_3$.” into the **Large non-reciprocal resistance in MnBi_2Te_4** (page 4 of the main text).

Comment 2.2: As was done in the previous study on Cr-doped $(\text{Bi,Sb})_2\text{Te}_3$ [Fig. 3, Nature Nanotechnology 15, 831 (2020)], the authors need to investigate how the magnetic-field-direction dependence of the non-reciprocal resistance looks like and to compare it with that of the Hall resistance. I think that these data are useful for explaining better the gate dependence and identifying what the underlying mechanism is. I am also curious how fundamentally different the suggested mechanism of non-reciprocity in this study is from the interplay between chiral edge state and surface state proposed previously [Nature Nanotechnology 15, 831 (2020)].

Response #2.2: We sincerely thank the reviewer for the insightful suggestion. Following the reviewer’s suggestion, we performed the magnetic-field-direction dependent measurements for a 5-SL MnBi_2Te_4 (Device 4). Figure R2.1 shows the schematic structure of the MnBi_2Te_4 device. For magnetic-field-direction dependent measurements, the 7 T magnetic field is rotated in the x - z plane. We first measure the Hall resistance under the out-of-plane magnetic field at $V_b = 0$ V and $V_b = -20$ V, respectively. The results are shown in Figure R2.1 b. The sample is hole doped at $V_b = 0$ V, and applying a negative bottom gate voltage increase the hole density in the sample. Then we measure the Hall resistance and non-reciprocal resistance when rotating the magnetic field. We show the results in Figure R2.1 (c) to (f). At $V_b = 0$ V, we observe a dip of $R_R^{2\omega}$ when the R_{xy} reaches the maximum value under an out-of-plane magnetic field. However, this feature disappears at $V_b = -20$ V, where $R_R^{2\omega}$ increases monotonically as R_{xy} increasing. This behavior agrees with previous reported non-reciprocal charge

transport in Cr-doped $(\text{Bi,Sb})_2\text{Te}_3$ and confirms that the non-reciprocal resistance in MnBi_2Te_4 originates from the interplay between chiral edge states and bulk states.

Figure R2.1: Magnetic-field-direction dependent measurements of a 5-SL MnBi_2Te_4 (Device 4).

All measurements are performed at 10 K with $I^{RMS} = 3 \mu\text{A}$. **a**, The schematic illustration of magnetic-field-direction dependent measurements. **b**, Hall resistance at $V_b = 0 \text{ V}$ and $V_b = -20 \text{ V}$. **c** and **d**, magnetic-field-direction dependent Hall resistance and non-reciprocal resistance at $V_b = 0 \text{ V}$ under -7 T . **e** and **f**, magnetic-field-direction dependent Hall resistance and non-reciprocal resistance $V_b = -20 \text{ V}$ under -7 T .

Non-reciprocal charge transport is enabled in quantum anomalous Hall systems, originating from the interplay between chiral edge states and bulk states. In this sense, non-reciprocal charge transports in MnBi_2Te_4 and Cr-doped $(\text{Bi,Sb})_2\text{Te}_3$ are similar. However, MnBi_2Te_4 shows some unique features, which is different from that in Cr-doped $(\text{Bi,Sb})_2\text{Te}_3$. We summarized the difference of non-reciprocal charge transports between MnBi_2Te_4 and Cr-doped $(\text{Bi,Sb})_2\text{Te}_3$:

- (1) Multiple non-reciprocal resistance states originating from the antiferromagnetic interlayer coupling. MnBi_2Te_4 is an A-type antiferromagnet, where magnetic moments are ferromagnetically coupled within the SL and antiferromagnetically coupled between two neighboring SLs. For a 5-SL MnBi_2Te_4 , applying an out-of-plane magnetic field results in various magnetic states, such as $\uparrow\uparrow\uparrow\downarrow\downarrow$, $\uparrow\downarrow\downarrow\downarrow\downarrow$, $\uparrow\uparrow\downarrow\downarrow\downarrow$, $\downarrow\downarrow\downarrow\downarrow\downarrow$, $\uparrow\uparrow\uparrow\uparrow$, and $\downarrow\downarrow\downarrow\downarrow$. Tuning the magnetization of the sample effectively modifies bulk-to-edge conduction ratio, manifesting in the Hall resistance plateaus and non-reciprocal resistance plateaus.
- (2) Layer-number manipulation of the non-reciprocal charge transport. The antiferromagnetic interlayer coupling makes even-SL and odd-SL MnBi_2Te_4 magnetically different under zero magnetic field. Without external magnetic field, odd-SL-number MnBi_2Te_4 is an uncompensated antiferromagnet with nonzero net magnetic moments, and even-SL-number MnBi_2Te_4 is a fully compensated antiferromagnet with zero net magnetic moments. Because of the different magnetic states, 5-SL samples show nonzero non-reciprocal resistance but 4-SL samples show almost vanished non-reciprocal resistance without magnetic field.

In summary, we ascribe the non-reciprocal resistance in MnBi_2Te_4 to the interplay between chiral edge states and bulk states. The van der Waals nature and the A-type antiferromagnetic structure make MnBi_2Te_4 unique, and thus we are able to manipulate the non-reciprocal charge transport of MnBi_2Te_4 by magnetic field, edge positions, gate voltage, and stacking sequence. We hope the reviewer found our reply thorough and reasonable.

Following reviewer's comment, we have included "Due to the ferromagnetic intralayer coupling and antiferromagnetic interlayer coupling, MnBi_2Te_4 also hosts rich magnetic phases, including fully compensated antiferromagnetic, uncompensated antiferromagnetic and spin-aligned ferromagnetic states. External magnetic field turns out to be an effective tool to manipulate such magnetic states as well as the charge transport." into the **Introduction of the main text (page 2)**, and "Further study about magnetic-field-direction dependent non-reciprocal charge transport is shown in Fig. S13. A dip of $R_R^{2\omega}$ when the R_{xy} reaches the maximum value under an out-of-plane magnetic field is observed in Device 4, indicating that the coexistence of chiral edge transport and bulk transport plays an important role in the non-reciprocal charge transport." into **Gate tunability of non-reciprocal charge transport of the main text (page 9)**.

Comment #2.3: Although the gate voltage does not switch the $\uparrow\downarrow\uparrow\downarrow$ to the $\downarrow\uparrow\downarrow\uparrow$ state (Fig. 4), there does exist a strong suppression of perpendicular magnetic anisotropy (by a factor of around 5, Fig. S8) when applying the gate voltage from -5 to +4 V. I agree with the authors that the sign change of the non-reciprocal resistance comes from the change of Fermi energy (relative to CNL). Yet, the gate-voltage-dependent magnetic anisotropy can also contribute to the magnitude. I think that the authors should clarify this point to avoid confusion.

Response #2.3: Thank you for such a great question. We totally agree with the reviewer that gate-voltage-dependent magnetic anisotropy should be seriously considered, since the non-reciprocal resistance in MnBi_2Te_4 is closely related to magnetic states. To answer this question, we studied the gate-voltage-dependent magnetization in MnBi_2Te_4 by optical magnetic circular dichroism (MCD). We would like to note that the Hall resistance in MnBi_2Te_4 is originated from both magnetism and band topology. Optical methods, such as MCD, can directly probe the magnetism in magnetic materials. We show MCD results for a 5-SL MnBi_2Te_4 at various gate voltage in Figure R2.2.

Figure R3.2: Gate voltage independent magnetization. **a**, Gate voltage dependent resistance of 5-SL MnBi_2Te_4 under zero magnetic field. The resistance shows a peak at the charge neutrality point. **b**, Optical MCD setup. The wavelength of the excitation laser is a 632.8 nm and the laser power is around 1 μW . **c**, Optical MCD at gate voltage of -14 V, -9.3V and 0 V.

We first determine the charge neutrality point (CNP) by measuring gate voltage dependent resistance under zero magnetic field. The resistance shows a peak at the CNP. Then we measure the optical MCD using the optical setup shown in Figure R3.2 b. The magnetic field is swept from -1 T to 1 T. We find the MCD hysteresis loops are independent to the gate voltage, indicating that gate voltage does not change the magnetization.

Based on the above discussions, the shrinking non-reciprocal resistance hysteresis loops at some gate voltages originate from the doping level change, during which the magnetization is unchanged. We thank the reviewer again for such a great suggestion, which helps a lot to improve our article.

Following reviewer's comment, we have now included "Besides, optical magnetic circular dichroism (MCD) reveals the magnetization is gate voltage independent in the field regime of -1 T to 1 T." into the **Gate tunability of non-reciprocal charge transport** of the main text (page 7), and included Figure R3.2 into the SI.

I hope it helps improve the overall quality of the paper to be more suited for the high-impact journal of Nature Communications.

Reviewer #3 (Remarks to the Author):

The manuscript entitled “Controlled large non-reciprocal charge transport in an intrinsic magnetic topological insulator MnBi_2Te_4 ” by Z. Zhang et al., described the nonreciprocal transport in magnetic topological insulator MnBi_2Te_4 . They reported gate- and magnetic-field-tunable nonreciprocal resistance. The authors attributed it to the hybridization between chiral edge state and trivial edge state. Data are nicely shown. However, there are a few issues which need to be addressed in this manuscript before it is suitable to be published in Nature Communications.

Comment #3.1: In the previous studies, nonreciprocal transport in MnBi_2Te_4 (C. Ye et al., Nano letters 22, 1366-1373 (2022)) and gate-controlled nonreciprocity in magnetic topological insulator (K. Yasuda et al., Physical Review Letters 117, 127202 (2016) and K. Yasuda et al., Nature Nanotechnology 15, 831-835 (2020)) has been already reported. What is the novelty of this paper?

Response #3.1: We sincerely thank the reviewer for such a great question. We are grateful to have an opportunity to discuss the novelty of our work. Ref [K. Yasuda et al., Physical Review Letters 117, 127202 (2016)] is a CBST/BST QAH/TI structure. Ref [C. Ye et al., Nano letters 22, 1366-1373 (2022)] is an MBT/Pt magnet/heavy metal structure. Non-reciprocity happens at the interface and chiral edge transport is not developed. In contrast, the development of chiral edge transport is crucial for our work and Ref [K. Yasuda et al., Nature Nanotechnology 15, 831-835 (2020)].

A unique property of quantum anomalous Hall materials is the conducting edge with the insulating bulk originating from the intertwined magnetism and band topology. This scenario supports the non-reciprocal charge transport when charge transport is in the dissipative regime, because the chiral edge transport breaks the inversion symmetry and the magnetic moment breaks the time-reversal symmetry on the edges. In this sense, MnBi_2Te_4 and Cr-doped $(\text{Bi,Sb})_2\text{Te}_3$ fall into similar material systems. In fact, MnBi_2Te_4 is a new member of the quantum anomalous Hall family and many fundamental questions remain. It is worthy to understand the charge transport in MnBi_2Te_4 , and study potential applications regarding following aspects:

- (1) Intrinsic magnetic topological insulator: To achieve quantum anomalous Hall effect, the ratio of elements in Cr-doped $(\text{Bi,Sb})_2\text{Te}_3$ needs to be precisely controlled, which is technically difficult. Meanwhile, the material quality is fundamentally limited due to magnetic dopants, since they act as impurities. However, this obstacle is removed in the intrinsic magnetic topological insulator MnBi_2Te_4 , which ensures the high material quality.
- (2) Multiple magnetic states resulting from the antiferromagnetic interlayer coupling: MnBi_2Te_4 is an A-type antiferromagnet. Within the SL, magnetic moments are ferromagnetically coupled. Magnetic moments in two neighboring SL are antiferromagnetically coupled. Such magnetic structures provide us with new degree of freedoms to manipulate the charge transport. First, without external magnetic field, odd-SL-number MnBi_2Te_4 is an uncompensated antiferromagnet with nonzero net magnetic moments, and even-SL-number MnBi_2Te_4 is a fully compensated antiferromagnet with zero net magnetic moments. Second, applying external magnetic field to multi-SL-number MnBi_2Te_4 aligns magnetic moments through spin-flip and/or spin-flop transitions. In contrast to ferromagnetic Cr-doped $(\text{Bi,Sb})_2\text{Te}_3$, magnetic phases of antiferromagnetic MnBi_2Te_4 are much richer, and thus host new possibilities to manipulate the charge transport in different magnetic states.
- (3) Capability of building van der Waals spintronics: While designing spintronic devices, van der Waals heterostructures are attractive. The atomically flat surface and the sharp interface are crucial to achieve high performance. The van-der-Waals nature of Mn_2BiTe_4 enables it to be combined with other van-der-Waals materials and achieve new device functions.

Based on above discussions, we believe our work provides new insights into the charge transport in MnBi_2Te_4 . We hope the reviewer found our reply reasonable and thorough.

Comment #3.2: Why hysteresis behavior in nonreciprocal signals at $V_g = -1$ (only R) and 5 V disappears (Fig. 4)? In that condition, hysteresis in R_{yx} is observed? For the clear understanding of the magnetic properties, you can show R_{yx} simultaneously in Fig.4. BTW, are there intermediate state in 4SL samples such as $\uparrow\uparrow\downarrow\downarrow$ or $\downarrow\downarrow\uparrow\uparrow$?

Response #3.2: We thank the reviewer for the careful reading and great questions. We totally agree for the reviewer that a clear understanding of magnetic states is crucial in our experiments. We would like to answer reviewer's questions point by point in the following:

- (1) In our gate dependent measurements, we found non-reciprocal resistance hysteresis loops shrink, or disappear, at some gate voltages. Here, we show the reciprocal resistance $R_R^{2\omega}$ (0 T) as a function of gate voltage in Figure R3.1a.

Figure R3.1: Gate dependent non-reciprocal resistance under zero magnetic field. The magnetic state is $\downarrow\downarrow\uparrow\uparrow$. **a.** and **b** are measured at right edge and left edge, respectively.

The shrinking non-reciprocity happens at three gate regime, $V_g = -5$ V, $V_g = -1$ V and $V_g = 5$ V. At $V_g = -5$ V and $V_g = 5$ V, the sample is heavily hole/electron doped. In this condition, metallic bulk dominates the charge transport, leading to conserved inversion symmetry, and thus non-reciprocity becomes weak. At $V_g = -1$ V, the $R_R^{2\omega}$ (0 T) undergoes a sign change from positive to negative, which results from the flipping order of Fermi velocity asymmetry between conduction and valence bands.

A slight difference between the magnitude of $R_R^{2\omega}$ (0 T) and $R_L^{2\omega}$ (0 T) is observed, which might originate from the asymmetric electrodes. Therefore, at $V_g = -1$ V, $R_R^{2\omega}$ (0 T) almost vanishes but $R_L^{2\omega}$ (0 T) shows a finite value.

- (2) To understand the relation between magnetic states and non-reciprocal charge transport. We show non-reciprocal resistance $R_L^{2\omega}$ and Hall resistance R_{xy} of Device 2 simultaneously in Figure R3.2.

Figure R3.2: Gate tunability. a, Gate dependent non-reciprocal resistance. **b,** Gate dependent Hall resistance.

Figure R3.2b indicates that the anomalous Hall resistance is tunable upon applying gate voltage. However, in a magnetic topological insulator, the anomalous Hall resistance originates from the combined magnetization and topological band structure. By measuring Hall resistance, we are unable to confirm whether the magnetization is changed upon applying gate voltage. To confirm the effect of gate voltage on magnetization, we perform optical magnetic circular dichroism (MCD) measurements for a 5-SL MnBi_2Te_4 at different gate voltages. As shown in Figure R3.3 a, the charge neutrality point is at $V_t = -9.3$ V, indicated by the resistance peak. We perform optical MCD measurements using the setup shown in Figure R3.3b. We show MCD loops at top gate voltage of -14 V, 9.3 V and 0 V in Figure R3.3c. The magnetic field is swept from -1 T to 1 T. Since the magnitude of MCD is proportional to the magnetization, unchanged MCD loops at different gate voltages reveal that the magnetization is gate independent in this field regime. Based on the above discussions, the shrinking non-reciprocal resistance hysteresis loops at some gate voltages originate from the doping level change, during which the magnetization is unchanged.

Figure R3.3: Gate voltage independent magnetization. **a**, Gate voltage dependent resistance of 5-SL MnBi_2Te_4 under zero magnetic field. The resistance shows a peak at the charge neutrality point. **b**, Optical MCD setup. The wavelength of the excitation laser is a 632.8 nm and the laser power is around $1 \mu\text{W}$. **c**, Optical MCD at gate voltage of -14 V, -9.3V and 0 V.

- (3) There are intermediate states in 4SL samples such as $\uparrow\downarrow\uparrow\uparrow$ or $\downarrow\downarrow\downarrow$. In Figure R3.4, we show non-reciprocal resistance of the 4-SL MnBi_2Te_4 as a function of magnetic field at $V_t = 0.5$ V. The magnetic states are labelled. The non-reciprocal resistance at magnetic states $\uparrow\downarrow\uparrow\uparrow$ or $\downarrow\downarrow\downarrow$ show different behaviors at different gate voltages. At $V_t = 3$ V as shown in the Figure 3b of the main text, non-reciprocal resistance vanishes. At $V_t = 0.5$ V, there is the finite non-reciprocity.

Figure R3.4: Non-reciprocal resistance of the 4-SL MnBi_2Te_4 as a function of magnetic field.

Following reviewer's suggestion, we have now revised Figure R3.2 in the SI, included "Besides, optical magnetic circular dichroism (MCD) reveals the magnetization is gate voltage independent in the field regime of -1 T to 1 T." into the **Gate tunability of non-reciprocal charge transport** (page 7 of the main text), and included Figure R3.3 into the SI.

Comment #3.3: The CNP is different between 5SL and 4SL samples (Fig. S3). Does this come from the sample dependence or layer number dependence? BTW, Fig. S3 is not mentioned in the main text nor Supplementary Materials.

Response #3.3: Thank you for the question. The doping levels of the pristine MnBi_2Te_4 devices that we have fabricated show sample dependence. Here we summarize carrier types in our four MnBi_2Te_4 devices in Table R3.1. We find the carrier type is independent to the SL number. We ascribe the difference of the carrier type between devices to the unintended doping during the fabrication process. The unintended doping might come from Al_2O_3 deposition process, PDMS films or the thermal release process.

Table R3.1:

Device	SL number	carrier type at $V_g = 0$
Device 1	5	electron
Device 2	5	hole
Device 3	4	hole
Device 4	5	hole
Device 5	5	electron

We have now included the description of Fig. S3 "For our Device 2 and Device 3, the nearly quantized Hall resistance of $0.977 h/e^2$ and $0.945 h/e^2$ have been achieved (Fig. S2 and Fig. S3)." into the **Chiral edge states revealed by quantum anomalous Hall effect** (page 4 of the main text).

Comment #3.4: There are several minor comments:

- (a) In the citation list, article numbers are sometimes wrong especially for Nature journals.
- (b) As for the nonreciprocal transport, there are other previous works:

R. Wakatsuki et al., Sci. Adv. 3 e160239 (2017).

P. He et al., PRL 120 266802 (2018)

J. Lustikova et al., Nature Communications 9 4922 (2018).

Y. M. Itahashi et al., Sci. Adv. 6 eaay9120 (2020).

Response #3.4: We thank the reviewer for the careful reading. We have revised the citation list and included all references.

Reviewers' Comments:

Reviewer #1:

Remarks to the Author:

The authors addressed my comments in full volume. Therefore, I am happy to recommend the publication in the present form.

Reviewer #2:

Remarks to the Author:

The new data and discussions have clearly improved the overall quality of this manuscript, and most of my concerns from the previous review have been properly addressed. Thus I would recommend the revised manuscript for publication in Nature Communications.

Reviewer #3:

Remarks to the Author:

Authors have provided rather detailed reply to the criticism. The data are nicely shown and results are interesting enough to be suitable for Nature Communications.

Response to Reviewers' comments

Reviewer #1 (Remarks to the Author):

The authors addressed my comments in full volume. Therefore, I am happy to recommend the publication in the present form.

Response #1: We sincerely thank the reviewer for helping us to improve the manuscript and the strong recommendation.

Reviewer #2 (Remarks to the Author):

The new data and discussions have clearly improved the overall quality of this manuscript, and most of my concerns from the previous review have been properly addressed. Thus I would recommend the revised manuscript for publication in Nature Communications.

Response #2: We are grateful for the reviewer's support. We sincerely thank the reviewer for the constructive comments and suggestions.

Reviewer #3 (Remarks to the Author):

Authors have provided rather detailed reply to the criticism. The data are nicely shown and results are interesting enough to be suitable for Nature Communications.

Response #3: We sincerely thank the reviewer for the careful reading and the strong support, which are important for us to improve the manuscript.